# Novel Subperiosteal Device Geometry and Investigation of Efficacy on Surrounding Bone Formation and Bone-Bonding Strength

**DOI:** 10.3390/bioengineering11111122

**Published:** 2024-11-07

**Authors:** Yoshiya Kaisaka, Masayoshi Uezono, Masaki Inoue, Kazuo Takakuda, Keiji Moriyama

**Affiliations:** 1Department of Maxillofacial Orthognathics, Graduate School of Medical and Dental Sciences, Institute of Science Tokyo, 1-5-45, Yushima, Bunkyo-ku, Tokyo 113-8549, Japan; y-kaisaka.mort@tmd.ac.jp (Y.K.); msk.i.kdu.4.2@gmail.com (M.I.);; 2Laboratory for Biomaterials and Bioengineering, Institute of Science Tokyo, 2-3-10 Kanda-Surugadai, Chiyoda-ku, Tokyo 101-0062, Japan; tkkdmech@tmd.ac.jp

**Keywords:** subperiosteal device, device geometry, bone formation, bone-bonding strength

## Abstract

To develop a safer bone-bonding device that promotes early osseointegration with cortical bone perforation, novel subperiosteal device geometries were proposed and evaluated for their ability to facilitate surrounding bone formation and enhance bone-bonding strength. This study used animal experiments and mechanical testing to assess the performance of these designs. The experimental device consisted of two main components: a rounded rectangular plate and a centrally positioned cylinder. To promote the recruitment of bone-marrow-derived factors, slits were incorporated into the cylinder, and a center hole was created directly above it. Four device variations, differing by the presence or absence of the slits and center hole, were fabricated and then subjected to tensile tests for mechanical property evaluation. In the animal experiments, the devices were bilaterally placed on rat tibiae, and after four weeks, bone-bonding strength tests were performed. Additionally, micro-computed tomography and histological analysis of undecalcified sections were conducted. All devices demonstrated early osseointegration, and geometric design differences, specifically the presence or absence of the slits and center hole, significantly affected the mechanical properties and bone induction. However, no significant differences in bone-bonding strength were detected. These findings suggest that the newly formed bone inside the slits and center hole contributes to the reinforcement of the device.

## 1. Introduction

Recently, numerous bone-bonding devices have been clinically utilized in oral and craniomaxillofacial surgery [1,2,3]. Most of these devices are drilled deeply into the craniomaxillofacial bones, which contain critical structures such as nerves, blood vessels, and teeth, posing a significant risk of tissue damage during placement [4,5,6]. To mitigate this risk, a subperiosteal device was developed that can be placed without the need for drilling [7]. This device, shaped like a disk with a diameter of 6.0 mm, is positioned under the periosteum and becomes biologically fixed to the bone through circumferential bone formation. Although this approach avoids the risks associated with deep drilling, it requires extended periods for osseointegration [8,9].

To reduce the osseointegration period, various strategies have been explored, including surface modification of the devices [10,11,12], the application of cytokines [13,14,15], and the use of drugs [16,17,18]. However, these techniques have encountered challenges in clinical application, such as systemic adverse effects and inconsistent efficacy [19,20]. On the other hand, cortical bone perforation is believed to effectively promote new bone formation [21]. Inspired by this concept, a novel subperiosteal device geometry was developed to achieve early osseointegration with cortical bone perforation instead of deep drilling during placement. Therefore, this study investigated the efficacy of the proposed subperiosteal device geometry in promoting surrounding bone formation and enhancing bone-bonding strength.

## 2. Materials and Methods

### 2.1. Preparation of the Specimen

All experimental subperiosteal devices were manufactured by a medical device company (Takashima Sangyo Co., Ltd., Nagano, Japan) with a general tolerance of less than ±0.05 mm. The devices were machined from grade II pure titanium blocks using a precision machining center. As illustrated in Figure 1A, each subperiosteal device consists of two main components: a rounded rectangular plate part (1.5 mm × 6.0 mm × 0.25 mm), designed to rest on the bone surface, and a centrally positioned cylinder (1.5 mm outer diameter, 1.0 mm inner diameter, and 1.0 mm height), intended for insertion into a pre-drilled hole in the cortical bone. This design is optimized to fit on the bone surface, promoting new bone formation around the device while ensuring secure attachment to the bone.

Bone-marrow-derived factors promote new bone formation when the cortical bone is damaged [21]. To facilitate the recruitment of these factors, slits were incorporated into the cylinder, and a center hole was created directly above it. To evaluate the effectiveness of these design features, four types of devices were fabricated, differing by the presence or absence of the center hole on the top of the cylinder (H+/H−) and the slits in the cylinder (S+/S−) (Figure 1C). The S−/H− group plays the role of control. A total of 14 devices were tested in each group.

### 2.2. Testing the Mechanical Properties of the Specimen

Tensile tests were performed using a universal mechanical tester (AG-X; Shimadzu, Kyoto, Japan) (*n* = 4) to evaluate the mechanical properties of the specimens. Figure 2A shows the procedure for testing the mechanical properties of the specimens. A stainless-steel rod (16 mm × 16 mm × 120 mm) was prepared with a drilled hole (1.5 mm in diameter and 1.0 mm in depth) located 1 mm from the upper end. The cylinder of the device was inserted into the drilled hole, and the lower one-third of the plate was stabilized using an auto-polymerizing resin (OSTRON II; GC, Tokyo, Japan). Then, a stainless-steel rod was attached to the lower vise of the mechanical tester, ensuring the load direction and the longitudinal axis of the device were aligned parallel.

For the mechanical test, a 0.38 mm diameter strand wire (wild cat wire; TOMY International, Tokyo, Japan) was threaded through one of the wire holes at both ends of the plate, tied, and secured to the upper vise. A tensile force was applied at a crosshead speed of 1 mm/min until the device failed at the rod. The force and displacement values were recorded throughout the test, and a force–displacement diagram was constructed. The point at which the force–displacement diagram deviated from the approximate curve in the elastic range was defined as the plastic deformation point, which was defined as the strength of the device.

### 2.3. Animal Experiment

Ethical approval for this study was obtained from the Institutional Animal Care and Use Committee of Institute of Science Tokyo, Tokyo, Japan (approval number A2017-298C). Twenty male Sprague-Dawley rats weighing 420–440 g and aged twelve weeks were used for the animal experiments, and the devices were attached to the tibiae of the rats bilaterally. Animals were fed a standard diet ad libitum and had unlimited access to water. Four types of devices were randomly selected and applied for each rat.

Prior to the experiments, the devices underwent ultrasonic cleaning in a detergent solution, followed by cleaning in acetone, ethanol, and deionized water. They were then irradiated with a UV/O2 cleaner (ASM401N; Asumi Giken Tokyo, Japan) for 2 min. The animals were anesthetized with a combination of midazolam (0.3 mg/kg BW; Dormicum; Sandoz, Tokyo, Japan), medetomidine (0.3 mg/kg BW; Domitor; Pfizer Animal Health, Exton, PA, USA), and butorphanol tartrate (0.3 mg/kg BW; Vetorphale; Meiji Seika Pharma, Tokyo, Japan) (MMB), administrated via intramuscular injection.

The area below the knee was shaved, and the skin was disinfected and draped with povidone-iodine (Isodine; Meiji, Tokyo, Japan). A midline incision was made along the tibia, followed by the avulsion of subcutaneous tissue and exposure of the periosteum. A 2.0 mm transverse periosteal incision was performed 3.0 mm distal to the knee joint, along with a 10 mm longitudinal incision along the medial lateral surface of the tibia. The medial side of the tibia was exposed using a small periosteal elevator, and a 1.4 mm pre-drilled hole was created 10 mm away from the knee joint to penetrate the cortical bone.

The cylinder of the device was placed in the pre-drilled hole with the long axis of the plate aligned parallel to the long axis of the tibia. The periosteum and skin flaps were closed and sutured with 5-0 synthetic absorbable sutures (Vicryl; Johnson & Johnson, New Brunswick, NJ, USA). Post-surgery, each rat was administered a dose of the antibiotic enrofloxacin (10 mg/kg BW; Baytril; Bayer AG, Nordrhein-Westfalen, Germany) and the anesthetic antagonist atipamezole hydrochloride (subcutaneous 0.3 mg/kg BW; Antisedan; Orion, Espoo, Finland) immediately after surgery to aid in recovery.

### 2.4. Evaluation

Four weeks after surgery, the experimental animals were euthanized using carbon dioxide, and the devices were carefully harvested together with the surrounding tibial bone for further analysis. The harvested samples were randomly divided into two distinct groups for different evaluation purposes: one group was preserved in frozen physiologic saline for mechanical testing (*n* = 20), while the other group was immersed in 70% ethanol for subsequent microcomputed tomography (μCT) and histologic examination (*n* = 20).

#### 2.4.1. Bone-Bonding Strength

Bone-bonding strength tests were performed within 12 h of harvesting the samples. The specimens were subjected to mechanical testing using a universal mechanical tester (AG-X; Shimadzu, Kyoto, Japan) at a crosshead speed of 1 mm/min until failure. Figure 2B illustrates the mechanical testing procedure. To prepare the sample, the tibial bone was cut off at 1.0 mm proximal to the plate and was carefully trimmed to expose a wire hole on one side of the plate, which was connected to the upper vise of the mechanical tester through the strand wire. Throughout the test, force and displacement values were recorded continuously until the device failed at the tibia. The bone-bonding strength was defined as the peak force observed on the force–displacement diagram, representing the maximum load the device could withstand before detaching from the tibia.

#### 2.4.2. μCT Image Analysis

To evaluate bone formation around the slits and center hole of each device, μCT images were obtained for all harvested samples within 6 h of collection using a μCT system (InspeXio SMX-100CT; Shimadzu, Kyoto, Japan) with a tube voltage of 45 kV, tube current of 30 μA, and voxel size of 20 μm. Three-dimensional (3D) images were reconstructed and analyzed using image analysis software (3D-Bon version 10.01; Ratoc, Tokyo, Japan). The threshold value for detecting newly formed bone was set at a density range of 900–1300 mg/cm^3^.

The region of interest (ROI) was defined as the cylindrical volume extending upward from the inside of the cylindrical part of the device. For devices with slits, the volume inside the slits was also included in the ROI. To reduce the influence of partial volume effect, four voxels surrounding the device were excluded from the analysis for each sample. As shown in Figure 3, Newly formed bone was segmented and color-coded into four distinct regions: Region 1 (red: above the center hole), Region 2 (green: inside the center hole), Region 3 (yellow: inside the cylinder), and Region 4 (blue: inside the slits).

For Regions 1–3, 3D images were evaluated from multiple perspectives, including top, bottom, and side views. The bone volume to tissue volume ratio (BV/TV) was then calculated for each region. In samples with slits, cross-sectional images were reconstructed at two levels: the upper level (corresponding to the top of the cylindrical part and the cortical bone area) and the middle level (corresponding to the middle of the cylindrical part and the cancellous bone area) to further investigate bone formation in these specific levels.

#### 2.4.3. Histological Observation

The samples were dehydrated with ethanol and infiltrated with a light-cured resin (Technovit 7200VLC; KULZER, Wehrheim, Germany). The samples were then embedded in the same resins and polymerized by exposure to ultraviolet light for 2 days. Undecalcified histologic sections, approximately 100 μm thick, were prepared using a diamond disk microtome (SP1600; Leica, Wetzlar, Germany) with the sections taken from the center of the longitudinal axis of the plate.

The prepared sections were stained with alizarin red and examined under a light microscope. To evaluate the contact between bone and device, the bone contact ratio (BCR) was calculated. The method for measuring the BCR is illustrated in Figure 4. The bone contact lengths on the top and outer surfaces of the cylinder were measured on both sides. The BCR was determined as the ratio of the bone contact length to the total length of the top and outer surfaces of the cylinder. Image analysis software (ImageJ version 1.53k; National Institutes of Health, Bethesda, MD, USA) was used to analyze the acquired images.

### 2.5. Statistical Analysis

All measured data were statistically analyzed using multiple comparisons of Wilcoxon rank-sum tests in combination with the Holm correction. Statistical significance was assumed for *p* < 0.05, and all the statistical analyses were performed using “R” software version 4.2.0 “http://www.r-project.org/, (accessed on 22 April 2022)”.

## 3. Results

All rats survived the entire experimental period without any sign of infection.

### 3.1. Mechanical Properties of the Specimen

Figure 5A shows the strength of the device. Significant differences were observed, except between the S+/H− and S+/H+ groups (*p* < 0.05). The S−/H− group exhibited the highest values. The strength of the S− group was higher than that of the S+ group. Among the S− group, the H+ group showed significantly greater values than the H− group (*p* < 0.05). In contrast, among the S+ group, no significant difference was found between the H+ group and the H− group (*p* < 0.05).

### 3.2. Bone-Bonding Strength Tests

Mechanical tests were successfully completed for all samples, with failure occurring at the interface between the device and the tibia. Figure 5B shows the test results, where the S−/H− showed the highest value; however, no significant differences were found between the groups.

### 3.3. μCT Image Analysis

Figure 6A displays the reconstructed μCT image of four regions (Regions 1–4). In Region 1, new bone formation decreased in the H− group. Regarding Region 2, new bone formation was observed along the lateral surface in the S−/H+ group, and more newly formed bone was seen in the S+/H+ group compared to the S−/H+ group. In Region 3, bone formation was observed along the lateral surface of the cylinder in the S−/H−, S−/H+, and S+/H− groups. In the S+/H+ group, bone formation along the lateral surface of the cylinder was minimal. In both the S+/H−, S+/H+ groups, a bone bridge was observed in Region 4.

The conditions for bone formation inside the slit are shown in Figure 6B. At the upper level, dense bone formation was observed in both groups. However, at the middle level, bone formation was sparse, particularly in the S+/H+ group. Figure 6C presents the BV/TV values for the three regions (Regions 1–3). In the Region 1, there were no significant differences between the pairs (*p* < 0.05). In Region 2, significant differences were observed between the H+ and H− groups (*p* < 0.05). In Region 3, the S+/H− group showed the highest BV/TV value, with significant differences except between the S−/H− and S+/H+, and between S−/H+ and S+/H+ groups (*p* < 0.05).

### 3.4. Histological Observations

Undecalcified sections stained with alizarin red are shown in Figure 7A. In the H+ group, new bone covered the top of the cylinder, with S−/H+ showing a greater amount of new bone than S+/H+. Figure 7B shows the BCR values, where no significant differences were found between the pairs.

## 4. Discussion

The amount of bone formation inside the cylinder was greater in the S+ group than in the S− group. This could have been because osteogenic factors entered the cylinder from the surrounding bone tissue through the slits. In addition, bone bridge formation between the slits was observed in some samples of the S+ group. Albrektsson et al. [22] reported that defect repair in vivo takes place over the shortest distance, supporting the assumption that bone bridges formed to connect the slits over the shortest distance inside the cylinder corresponding to the bone defect. Similar findings were reported by Kim et al. [23], where a hollow implant with a spiral-shaped side opening facilitated new bone formation at the side opening and bone bridging inside the hollow part three weeks after healing.

From the reconstructed μCT images of the H− group, a large amount of bone formation was observed inside the cylinder. In contrast, the newly formed bone inside the cylinder was sparsely observed in the H+ group. The reason for this result could be considered to explain by the accumulation of cytokines. The presence or absence of the center hole may affect cytokine concentrations inside of the cylinder part; however, more research is necessary to verify the involvement of cytokine.

Mechanical strength tests were performed with a stainless-steel rod instead of a tibia to investigate the mechanical properties of the specimen without the influence of new bone formation around the device. As the strength of the stainless-steel rod is significantly higher than that of the device, it allowed the estimation of the starting point of plastic deformation. The results indicated that the S− group had significantly higher values than the S+ group, demonstrating fact that the slits significantly reduced the mechanical properties of the specimen. The impact of the center hole on the mechanical properties of the specimens depended on the slit conditions. In the S+ condition, the center hole had no effect on the specimen’s mechanical properties, whereas in the S− condition, it significantly reduced the mechanical properties of the specimen.

The results of the mechanical properties suggested that the center hole or slits would make the devices more deformable. Deformable device geometries are expected to result in lower bone-bonding strength [24]. However, in this study, bone-bonding strength did not show the same trends as the mechanical properties. In the S+/H− and S+/H+ groups, the cylinder’s deformation due to the slits was mitigated by bone formation inside the slits, particularly at the cortical bone level, which increased bone-bonding strength. Figure 6B shows that dense bone formation was observed inside the slit at the cortical bone level in both groups. Since stress concentrates around the cortical bone when a lateral force is applied [25,26], new bone formation at the cortical level helped suppress cylinder deformation, and as a result, the bone-bonding strength values of the S+/H− and S+/H+ groups were not significantly lower than that of the S−/H− groups. In the S−/H+ group, although the center hole reduced the mechanical properties of the specimen, new bone formation inside and above the center hole likely increased bone-bonding strength, resulting in no significant difference being observed between the S−/H+ and S−/H− groups.

Our proposed subperiosteal device is intended for use as an orthodontic anchorage device for traction of teeth and jawbone [3,7]. In this study, the bone-bonding strength test focused on horizontal loading along the longitudinal axis of the device, which is the most commonly expected clinical load. However, in practice, traction loads are applied in multiple directions, so other load directions should be considered. When the device is loaded perpendicular to the bone surface, newly formed bone above or in contact with the device can help prevent detachment. Histological examinations and μCT observations showed that a greater volume of newly formed bone on the top of the device in the H+ group compared to the H− group. Additionally, in the S+ group, the contact area for new bone formation inside the slits was larger than that of the S− group. Therefore, the S+/H+ configuration is likely the most suitable for an orthodontic anchorage device, as bone-bonding strength in this group does not depend on the direction of traction. As for bone-bonding strength, that of our experimental devices was approximately 40 N. Considering that the maximum orthodontic force is 5 N [27], the current bone-bonding strength should be sufficient for orthodontic application. However, since the horizontal pull-out strength of the commercially available anchorage screw is 166.3 N [27], additional ingenuity to improve bone-bonding strength has to be discussed.

One limitation of this study is the size of the device. The slits and center hole were designed to provide a pathway for osteogenic factors to reach the bone surface, but blood clots often block the center hole during device placement. Furthermore, BCR results were highly variable, with some samples showing no bone formation above the center hole, even under the H+ condition. These variations could be due to the size of the slits and center hole. In this study, the device was scaled for rats, and it is expected that clinical devices would be larger, potentially improving the function of the slits and center hole as pathways. Another limitation is the placement site. The craniofacial region is considered a good application for the new subperiosteal device because the bone in this region contains important tissues. The craniofacial region, particularly the cranium, maxilla, and mandible, which are formed by intramembranous ossification [28,29], may differ in bone formation from the results observed in this study using the tibia, which is formed by endochondral ossification. Further studies are needed to evaluate the performance of the new subperiosteal device in membranous bone.

## 5. Conclusions

Each experimental subperiosteal device successfully achieved osseointegration within four weeks of placement. The experimental subperiosteal devices showed considerable differences in mechanical properties and bone induction performances according to their geometrical designs, i.e., the slits significantly reduced the mechanical properties of the specimen, and in the S− condition, the center hole also significantly reduced them. However, there were no significant differences in bone-bonding strength, demonstrating the strengthening effects attributed to the newly formed bone inside the slits and center hole.

## Figures and Tables

**Figure 1 bioengineering-11-01122-f001:**
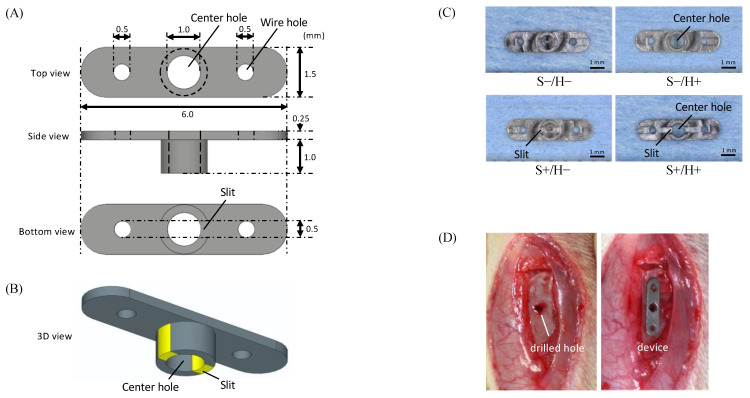
(**A**) Schematic representation of the experimental orthodontic anchorage device. The upper panel: a top view of the device. The middle panel: a side view. The lower panel: a bottom view. (**B**) Three-dimensional view of the experimental orthodontic anchorage device with the slits highlighted in yellow. (**C**) Images of the experimental subperiosteal devices viewed from the bottom with the slits highlighted in green. Bar = 1 mm. (**D**) Intraoperative images. The left panel: making pre-drilled hole. The right panel: placement of the specimen.

**Figure 2 bioengineering-11-01122-f002:**
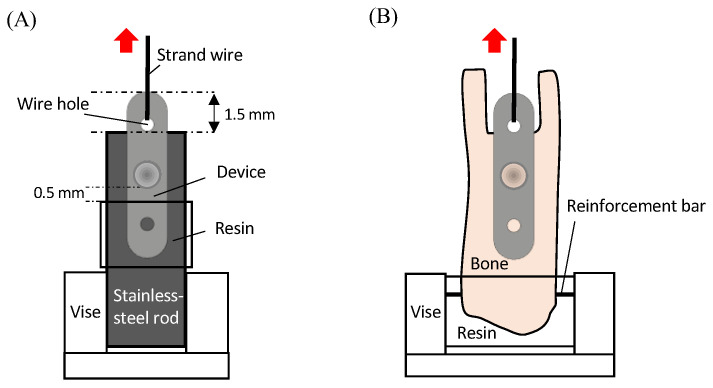
Schematic representation of the mechanical test, front view. Red arrows indicate the direction of traction. (**A**) Mechanical properties test of the specimen. (**B**) Bone-bonding strength test of the specimen.

**Figure 3 bioengineering-11-01122-f003:**
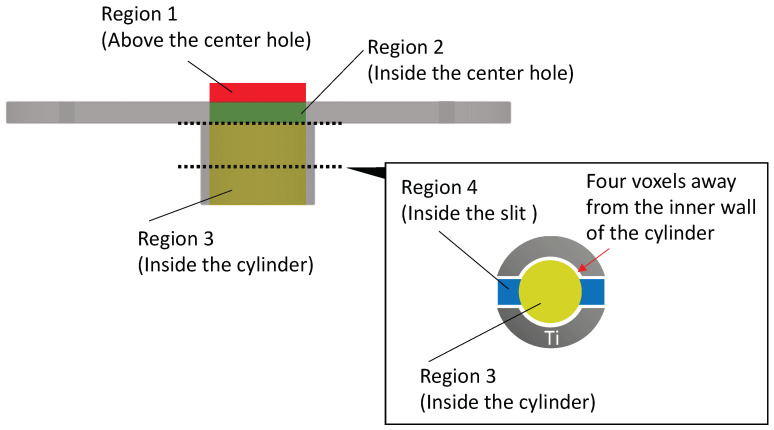
Region of interest (ROI) for μCT image analysis. A schematic side view of the device. The upper dotted line marks the boundary between the plate and the cylindrical parts, while the lower dotted line indicates the boundary at the center of the cylinder. The cross-sectional view is displayed in the rectangle at the lower-right. The red area represents the ROI above the center hole (Region 1), the green area represents the region of interest (ROI) inside the center hole (Region 2), the yellow area shows the ROI inside the cylinder (Region 3), and the blue area shows the ROI inside the slit (Region 4). Each ROI was 4 voxels away from the inner surface of the device.

**Figure 4 bioengineering-11-01122-f004:**
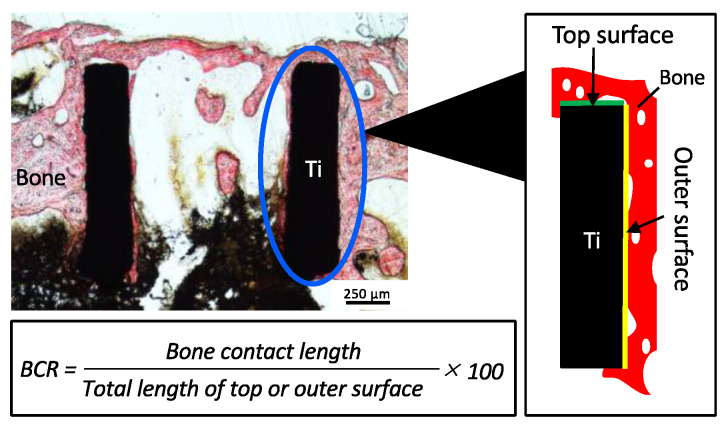
Bone contact ratio (BCR) measurements in histologic sections. The left panel displays sections stained with alizarin red, while the right panel provides a magnified schematic of the bone contact area. The black area represents the titanium device, and the red area indicates the newly formed bone. Green and yellow lines indicate the top and outer surfaces of the device, respectively. Bar = 250 μm.

**Figure 5 bioengineering-11-01122-f005:**
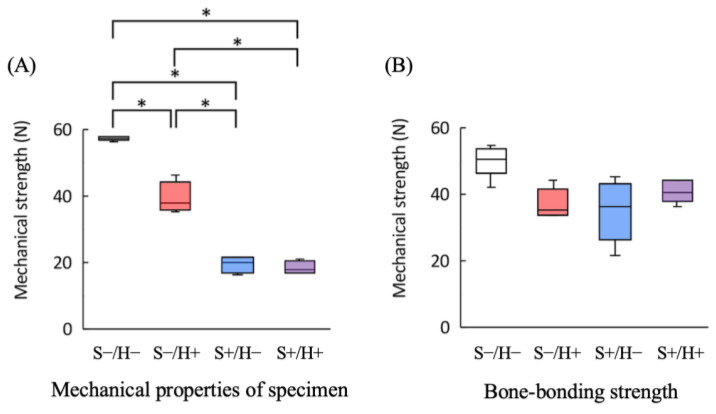
Box-and-whisker plot showing the result of the mechanical test. (**A**) Mechanical properties of the specimens (*n* = 4). The asterisk indicates a significant difference between the groups (*p* < 0.05). (**B**) Bone-bonding strength of the device (*n* = 5).

**Figure 6 bioengineering-11-01122-f006:**
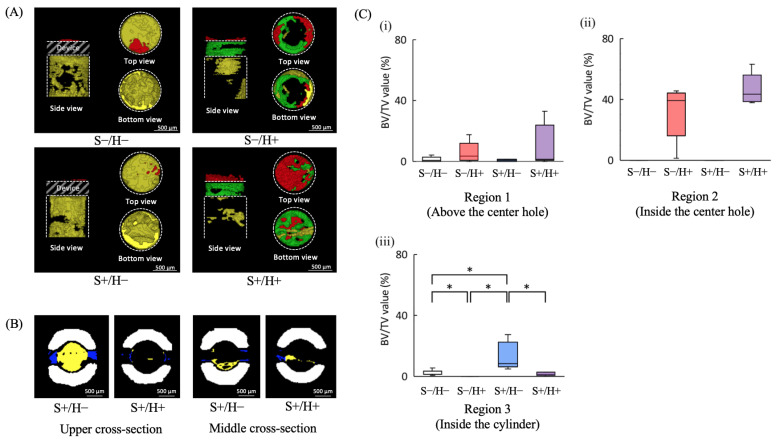
Results of μCT analysis. (**A**) Typical reconstructed 3D μCT images showing the areas inside and above the cylinder. The red area represents the region above the center hole (Region 1), the green area represents the region inside the center hole (Region 2), the yellow area indicates the region inside the cylinder (Region 3). The shaded area represents the device. Bar = 500 μm. (**B**) Reconstructed 3D μCT images from inside the slit. The left panel shows a horizontal cross-sectional image at the upper border of the cylinder, while the right panel shows a horizontal cross-sectional image at the middle of the long axis of the cylinder. The white area represents the titanium device, while the blue and yellow areas show new bone formation. Bar = 500 μm. (**C**) Box-and-whisker plot of BV/TV (*n* = 5). (**C-i**) BV/TV value above the center hole, (**C-ii**) BV/TV value inside the center hole, (**C-iii**) BV/TV value of inside the cylinder. The asterisk indicates significant differences between groups (*p* < 0.05).

**Figure 7 bioengineering-11-01122-f007:**
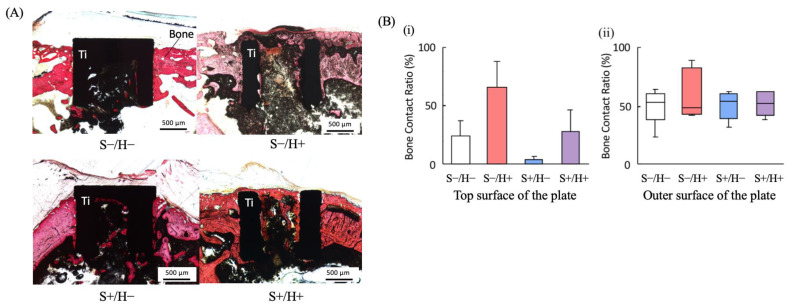
Results of histologic image analysis. (**A**) Light microscopic images showing the titanium device (black) and bone (red). Scale bar = 500 μm. (**B**) Box-and-whisker plot of the bone contact ratio (BCR) (*n* = 5). (**B-i**) BCR of the top surface of the plate. (**B-ii**) BCR of the outer surface of the plate.

## Data Availability

The data presented in this study are available on request from the corresponding author. The data are not publicly available due to patent issues.

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
