# Peer review of "Novel Subperiosteal Device Geometry and Investigation of Efficacy on Surrounding Bone Formation and Bone-Bonding Strength"

_bioengineering, 2024, doi:10.3390/bioengineering11111122_

Round 1

Reviewer 1 Report

Comments and Suggestions for Authors

This manuscript investigates the influence of two different modifications made on a subperiosteal device in the capacity of the device for bone integration and mechanical properties. Some concerns follow:

1.      The sentence in line 37 is confusing, reference 21 actually concludes:

…that cortical bone perforation favourably affects the amount of new bone formation in the grafted sites after 7 months of healing. Cortical bone perforation significantly increase number of new vessels (angiogenesis) of the regenerated bone and may provide some advantages without any serious negative effect…..

Then, What is the meaning of …minimal bone invasion….promote bone formation?

2.      The first paragraph of the discussion seems to over reach the experiments and results. Bone regeneration can and is actually followed through the expression of cytoquines as properly pointed with reference 22 but this is not what it is studied in this work, why focusing the discussion in this point? There is no possible conclusion, no cytoquines have been studied. All the first paragraph is out of focus and should be rewritten and maybe incorporated later in the section

3.      Line 232, The verb …suggesting … does not describe the results. It should be changed for a more direct verb

4.      Conclusions- line 273 the sentence ,….particularly the presence or absence  of slits and  a center hole… is too ambiguous for the conclusion of this work and not  coincident with the reported different influence of the slit and the hole on the mechanical properties of the device (fig 2b)

5.      Line 276 ….bone bonding strength is not affected by the direction of traction… This can be discussed but No mechanical test experiments on different directions have been reported to be able to conclude this.

Comments on the Quality of English Language

The meaning of some verbs used in the manuscript may not be the intended

Reviewer 2 Report

Comments and Suggestions for Authors

The paper considers a novel design of subperiosteal device, which attempts to eliminate the requirement for deep drilling whilst encouraging osteointegration through what is effectively surface modification of the device. The premise appears sound although the design of the device is not adequately described and it is not clear what the slits are or what exactly they are designed to achieve. A generally rigorous methodology is described with some specific points addressed below.

Line 37 – Minimal bone invasion believed to effectively promote new bone formation? The reference you refer here describes the positive influence that cortical bone perforation has on osseointegration – in other words greater invasion into the bone induces greater bone perforation I am therefore confused as to why it is proposed that minimal bone invasion (which I take to mean reduced bone invasion is suggested to be effective). I understand you are trying to differentiate from the deep drilling required by alternative devices, but I would suggest minimal bone invasion is not the best phrase to use.

Description of geometry in Figure 1 – it is unclear what the slits are – it doesn’t look like they are through thickness, do they sit at the bottom of the implant? The engineering drawings do not appear to be representative of the photographs shown in the same figureHow do the slits result in any invasion into the bone – is it just in effect providing porosity in the structure which numerous literature has shown to promote bony integration.

It would be highly beneficial to include a figure (ideally photographs) of the surgery and implant site.

All methodologies appear sound, processes correct and ethics requirements adhered to. How were the 20 animals divided into the various samples types were there any controls?

Characterisation of bone formation is comprehensive, however from the figures describing the mechanical testing it is not clear how shear forces were applied – it appears to be uniaxial tensile loading. ` Tensile loading is fine, but this should be correctly stated or nature of shear loading more clearly described.       

Whilst a thorough description of the mechanical testing is described within  the results the Discussion of mechanical testing Results are a bit confusing (from line 237):

S-/H- lower bone bonding strength than mechanical properties of specimen

S-/H+ almost identical value to the specimen’s mechanical properties

S+/H- and S+/H+ groups had high bone-bonding strength than specimen.

Therefore – no significant differences in bone bonding strength between the groups. The above statements do not seem to attest to this final conclusion.

 With no significant difference between any of the sample types it is not clear what the significance of this work is, more discussion comparing to existing devices would be beneficial.

Round 2

Reviewer 1 Report

Comments and Suggestions for Authors

The authors have reponded to all my former comments

Comments on the Quality of English Language

no further concerns